# Diversity in Cell Morphology, Composition, and Function among Adipose Depots in River Buffaloes

**DOI:** 10.3390/ijms24098410

**Published:** 2023-05-07

**Authors:** Xintong Yang, Ruirui Zhu, Ziyi Song, Deshun Shi, Jieping Huang

**Affiliations:** State Key Laboratory for Conservation and Utilization of Subtropical Agro-Bioresources, Guangxi Key Laboratory of Animal Breeding, Disease Control and Prevention, Guangxi University, Nanning 530005, China; tty970512@126.com (X.Y.); zhuruirui0116@126.com (R.Z.); ziyi.song@gxu.edu.cn (Z.S.)

**Keywords:** *Bubalus bubalis*, adipose tissue depot, cell size, composition, function, metabolism

## Abstract

Fat deposition is a significant economic trait in livestock animals. Adipose tissues (ATs) developed in subcutaneous and visceral depots are considered waste whereas those within muscle are highly valued. In river buffaloes, lipogenesis is highly active in subcutaneous (especially in the sternum subcutaneous) and visceral depots but not in muscle tissue. Revealing the features and functions of ATs in different depots is significant for the regulation of their development. Here, we characterize the cell size, composition, and function of six AT depots in river buffaloes. Our data support that the subcutaneous AT depots have a larger cell size than visceral AT depots, and the subcutaneous AT depots, especially the sternum subcutaneous AT, are mainly associated with the extracellular matrix whereas the visceral AT depots are mainly associated with immunity. We found that sternum subcutaneous AT is significantly different from ATs in other depots, due to the high unsaturated fatty acid content and the significant association with metabolic protection. The perirenal AT is more active in FA oxidation for energy supply. In addition, the expression of HOX paralogs supports the variable origins of ATs in different depots, indicating that the development of ATs in different depots is mediated by their progenitor cells. The present study enhances our understanding of the cellular and molecular features, metabolism, and origin of AT depots in buffaloes, which is significant for the regulation of fat deposition and provides new insights into the features of AT depots in multiple discrete locations.

## 1. Introduction

Fat deposition, namely adipose tissue (AT) development, is a vital economic trait in livestock animals. AT can develop in multiple anatomical depots, which mainly include subcutaneous fat (SCF), visceral fat (VCF), and intramuscular fat (IMF) in livestock animals. Among them, IMF is highly valued whereas others are considered waste [1]. Thus, a strategy on regulating AT development in specific depots is highly appreciated in livestock animals. Characterizing the differences of ATs in multiple discrete locations will contribute to optimizing fat deposition in livestock animals.

Buffaloes (*Bubalus bubalis*) play vital roles in the lives of tens of millions of people as a source of milk, meat, and draught power. Buffaloes are regarded as the most promising domestic animal for meat production [2]. Buffaloes are mainly divided into the river type and swamp type. Most buffaloes are of the river type for their high body size and good performance in milk production. In fact, lipogenesis is highly active in river buffaloes. For example, the subcutaneous AT is abundant and the milk fat content is very high in river buffaloes [3]. However, lipid accumulation in muscle is very limited in river buffaloes, which makes their meat quality very poor. To date, information on the development and regulatory mechanisms of ATs in different depots is limited in buffaloes.

The development of ATs is accompanied by the proliferation and adipogenic differentiation of preadipocytes. The migration and proliferation of preadipocytes are mainly completed during the embryonic stage. In the later stage of growth, preadipocytes are induced to adipogenic differentiation by an adequate energy supply, with lipid droplets accumulating in the cells and resulting in an increase in cell size. Thus, the cell size of adipocytes is an important indicator for evaluating the development of ATs in different depots [4,5,6]. The excess energy is mainly stored in mature adipocytes in the form of triglyceride (TG). A molecule of triglyceride is formed by one molecule of glycerol and three molecules of fatty acids. In the state of energy mobilization, triglycerides are decomposed into fatty acids, which are transported into mitochondria and undergo β-oxidation to release energy [7]. Fatty acid composition in different AT depots is variable [8,9], indicating that ATs in different depots have variable preferences for the absorption and synthesis of different types of fatty acids. Accordingly, gene expression profiling is different in different AT depots [4,6] as they play multiple roles in the health of animals [10,11]. Most of the existing results are based on studies in humans and model animals. The distribution of AT depots throughout the body in different species can be different [12]. In river buffaloes, lipogenesis is highly active. Specially, they have well-developed sternum subcutaneous AT that may be different from those in other depots.

In the present study, the cell size, TG and crude fat contents, main fatty-acid composition, and transcriptomes of ATs from six depots in river buffaloes, including three subcutaneous AT (SAT) depots (back subcutaneous AT, BSAT; sternum subcutaneous AT, SSAT; inguinal AT, IAT) and three visceral AT (VAT) depots (omental AT, OAT; pericardial AT, PCAT; perirenal AT, PRAT), were studied. We characterized the cellular and molecular properties and functions of ATs from discrete locations. The results will provide essential information for the development of diverse AT depots in buffaloes.

## 2. Results

### 2.1. Cellular Morphology

To reveal potential differences in cellular morphology between different AT depots (Figure 1a), H&E staining (Figure 1b) was performed for the six AT depots in the buffaloes. As shown in Figure 1b, complete cell morphologies were obtained for all depots except the SSAT depot (Figure 1b). Then, the number of cells per unit area instead of the cell area was analyzed. As shown in Figure 1c, the cell number per field in the PRAT depot was the highest, followed by the OAT depot. The cell numbers of the PCAT, IAT, SSAT, and BSAT depots were the lowest (Figure 1c). Therefore, the cell size in the PRAT depot was the smallest, followed by the OAT depot and then by the other four depots.

### 2.2. Triglyceride Content and Crude Fat Content

To reveal potential differences in components between different depots, both TG content and crude fat content were detected for all 36 AT samples. Interestingly, for each individual, the highest TG content was detected in the IAT depot (Figure 1d). The levels of TG in the IAT, BSAT, and OAT depots were widely distributed whereas those in the other three depots were stable. The SSAT had the lowest TG content (Figure 1d). By contrast, the lowest crude fat content was detected in the SSAT depot and the highest crude fat content was detected in the PCAT depot; other depots had similar crude fat content (Figure 1e). Thus, TG content was not consistent with crude fat content in AT depots.

### 2.3. Fatty Acid Composition

The main fatty acids, including 3 saturated fatty acids (SFAs) (stearic acid, myristic acid, and palmitic acid) and 4 unsaturated fatty acids (UFAs) (oleic acid, palmitoleic acid, linoleic acid, and α-linolenic acid), were detected for the 36 samples across the six AT depots. As shown in Figure 2a–c, stearic acid was the most abundant SFA in ATs. The highest stearic acid content was detected in the PRAT depot, followed by OAT and PCAT depots (Figure 2a). The SSAT depot had the lowest stearic acid content, followed by the BSAT depot (Figure 2a). The level of stearic acid content in the IAT depot was similar to that in the VAT depots. For the UFAs (Figure 2d–f), oleic acid was the most abundant one. The SAT depots had higher oleic acid and palmitoleic acid contents, especially in the SSAT (Figure 2d,e). The lowest palmitoleic acid content was detected in PRAT (a VAT depot); the highest level was detected in SSAT (a SAT depot) (Figure 2e). In total, the SFA contents in three VAT depots and the IAT depot were more than 50% and those in the other two SAT depots (BSAT and SSAT) were <50% (Figure 2h). Consistently, BSAT and SSAT depots had relatively high UFA contents (Figure 2i). In most AT depots, the UFA content was lower than the SFA content; however, in the SSAT depot, the pattern was the opposite (Figure 2j,k).

### 2.4. Overview of the RNA Sequencing

Since cell sizes and components varied across AT depots, RNA sequencing was performed to reveal potential molecular differences. In total, 36 ATs across the 6 depots were used for RNA sequencing. A total of 5.3 billion 100 nt pair-end clean reads were obtained (Appendix A). A river buffalo genome (assembled by our team, not published) was used and the mapping ratios were >96% (Appendix A). Approximately 23,000 transcripts were identified (average FPKM >0.1, Appendix A). The numbers of transcripts identified in different depots were similar (Figure 3a and Appendix A).

### 2.5. Functional Divergence between Subcutaneous and Visceral Depots

Based on all the identified transcripts, SATs were generally distinguished from VATs (Figure 3b,c). Expression analysis revealed 767 differentially expressed genes, with 228 genes upregulated in SATs and 644 genes upregulated in VATs (Figure 3d and Appendix A). Functional enrichment analysis indicated that the upregulated genes in SATs were mainly involved in the extracellular matrix (space/region) (Figure 3e and Appendix A). The genes upregulated in VATs were mainly involved in immunity (Figure 3f and Appendix A). In addition, we found that the 26 identified HOX paralogs (Appendix A) divided the 6 AT depots into 4 clusters (Figure 3g). The BSAT, IAT, and PRAT depots were clustered into one branch. Meanwhile, the OAT and SSAT depots clustered together, then they clustered with the other three depots (BSAT, IAT, and PRAT) (Figure 3g). The PCAT depot seemed to be different from the others (Figure 3g,h).

### 2.6. WGCNA Reveals the Functional Features of SSAT and PRAT Depots

To combine phenotypic differences with the internal molecular events, a WGCNA was performed by transcripts with an average FPKM >2 (approximately 10,100 transcripts). Four samples deviating from others were removed (Appendix A) and a total of thirty-two samples (Figure 4a) were used. Based on the scale-free topology model fit (Figure 4b), a soft threshold with 4 was used. All the transcripts were mainly assigned to 12 modules (Figure 4c). Then, correlation analysis between 11 phenotypic traits (BSAT depot, SSAT depot, IAT depot, OAT depot, PCAT depot, PRAT depot, SAT depots, VAT depots, TG content, SFA content, and UFA content) and 12 modules was performed (Figure 4d). A total of seven modules had a significantly high correlation with one or more phenotypic traits. Combined with functional enrichment analysis, two key points were found.

The first interesting point was that both green-yellow and yellow modules had a significant correlation with the SSAT depot, SAT depots, and UFA content. The module–trait relationships indicated that the green-yellow module, yellow module, and trait (SSAT or SAT or UFA) clustered into one branch (Appendix A). Thus, the 75 genes in the green-yellow module and the 183 genes in the yellow module were merged for further functional enrichment. Genes in green-yellow and yellow modules were mainly involved in categories related to the extracellular matrix (ECM), as well as collagen, a significant constituent of the ECM [13] (Figure 4e), which were classified as candidate categories for further analysis (Appendix A, labeled with red color).

The second key point was that three modules (purple, blue, and brown) had a significantly high correlation with the PRAT depot, VAT depots, and SFA content (Figure 4d). Functional enrichment showed that genes in the brown module were mainly associated with the tricarboxylic acid cycle, mitochondria, and ATP (Figure 4f; Appendix A, labeled with red color). Many categories related to FA beta-oxidation, degradation and metabolism, and mitochondria were enriched in the blue module (Figure 4g; Appendix A, labeled with red color). Thus, related categories were classified as candidate items used for further analysis (Appendix A, labeled with red color). No significant items were identified by the 76 genes in the purple module.

Turquoise had a significant correlation with VCF; few items associated with immunity were identified (Appendix A, marked with red color, 8/386). The pink module had a significantly high correlation with the PCAT depot, VAT depots, and SFA content; the red module was correlated with the PCAT depot (Figure 4d). However, functional enrichment identified only 1 significant item (GO:0005515~protein binding) by the 128 genes in the pink module and only 2 items (GO:0016324~apical plasma membrane and GO:0098793~presynapse) in the red module (Appendix A), which did not present valuable information for the PCAT depot, VAT depots, or SFA content. Thus, we did not pay further attention to the pink and red modules.

### 2.7. Screening and Expression of Core Genes

To reveal the key genes responsible for the ECM-associated regulatory network in green-yellow and yellow modules and for the regulatory network associated with FA degradation and metabolism in brown and blue modules, expressional profiles of genes involved in the candidate items (Appendix A) were analyzed across the six AT depots. Based on the RNA sequencing, most genes involved in the candidate items in green-yellow and yellow modules are upregulated in the SSAT depot, especially the *VCAN* gene (Figure 5a). A qRT-PCR analysis further confirmed the high expression of the top two genes (*VCAN* and *FMOD*) in the SSAT depot (Figure 5b,c). The *VCAN* gene is involved in the GO:0031012~extracellular matrix and GO:0005576~extracellular region and the *FMOD* gene is involved in GO:0030199~collagen fibril organization and GO:0005615~extracellular space (Appendix A). A co-expression network indicated that both *VCAN* and *FMOD* interact with many other genes associated with the ECM (Appendix A).

Similarly, most genes involved in the candidate items of the brown module have the highest expression levels in the PRAT depot, especially the HSPD1 and LDHB genes (Figure 5d). The high expression of *HSPD1* and *LDHB* in PRAT were further identified by qRT-PCR (Figure 5e,f). The *HSPD1* is involved in four categories of mitochondrial components and ATP while the *LDHB* gene is associated with mitochondrial inner membrane and metabolic pathways (Appendix A). The co-expression network indicated that *HSPD1* has complex interactions with many other genes involved in the mitochondrial inner membrane/matrix and ATP binding/enzymatic activity (Appendix A); the *LDHB* presents complex interactions with many genes involved in bta01100 metabolic pathways (Appendix A) and the GO:0005743~mitochondrial inner membrane (Appendix A). Meanwhile, *GPD1* and *HADHA*, which are involved in the blue module, have the highest expression in PRAT (Figure 5g). *GPD1* is involved in the bta00564: glycerophospholipid metabolism pathway and the *HADHA* gene is associated with nine categories, which mainly included FA metabolic (Appendix A). Both *GPD1* and *HADHA* present considerable associations with other genes in the corresponding items (Appendix A).

## 3. Discussion

ATs in different depots can be different in cell morphology, composition, and biological function. This study characterizes the cellular and molecular properties, as well as the function and metabolism features, of ATs from six discrete locations in river buffaloes, including three SAT depots and three VAT depots. Data in the present study support that (1) SAT depots have a larger cell volume than VAT depots, indicating that SAT is better developed than VAT; (2) the SSAT depot is significantly different from other AT depots in composition and function, which may be a specific depot in river buffaloes; (3) PRAT is more active in FA oxidation for energy supply; (4) SATs, especially the SSAT, is mainly associated with the ECM and VATs are mainly associated with immunity; (5) simply dividing ATs into the groups SAT and VAT is not enough to characterize existing variation.

It is known that adipose development can be affected by gender. In the present study, no significant differences were found in cell size or in FA and crude fat content between male and female individuals (data were not shown). This may be due to the small sample size (three male and three female individuals). Thus, the samples were not grouped according to gender. However, some interesting results were still revealed by ignoring gender in the present study.

### 3.1. SAT Depots Have a Larger Cell Volume Than VAT Depots

Generally, the development of AT after birth is mainly characterized by the increased volume of adipocytes. In other words, the volume of adipocytes indicates the level of maturity in ATs. In animals, AT preferentially deposits in subcutaneous depots then in visceral depots; intermuscular AT deposition is the last and the most difficult. Previous studies have shown that the cell volume of adipocytes in different depots can be variable in pigs [4,14] and cattle [6,15]. The SAT depots generally have larger adipocyte volumes than those in VAT depots [4,14]; the intermuscular AT has a smaller cell volume than those in other AT depots [4,6]. Similarly, in river buffaloes, our data showed that PRAT has the smallest adipocyte volume, followed by OAT, and the other four SAT depots have the largest adipocyte volumes (Figure 1c). These results indicate that excess energy is easier to store in the form of lipid droplets in SATs and PCAT than in PRAT and OAT, which was also confirmed by the following WGCNA and functional enrichment. PRAT had a significantly high correlation with FA beta-oxidation and degradation, tricarboxylic acid cycle, and energy metabolism (Figure 4d,g). In PRAT, more FAs and TGs are used to produce ATP, resulting in fewer lipid droplets. Thus, the adipocyte volume is smaller in PRAT. OAT also had a considerable positive association with the brown module whereas the other four AT depots show a negative association with the brown module (Figure 4d).

### 3.2. SSAT Is Significantly Different from ATs in Other Depots

Adipocytes are the main cellular component of AT and store energy in the form of TG droplets [16]. Our data showed that SAT depots, except for SSAT, have a higher TG content than the VAT depots (Figure 1d). Except for SSAT, ATs in other depots have similar crude fat content (Figure 1e). WGCNA reveals that SSAT is associated with the ECM (Figure 4d,e). Further expression profile analysis confirmed that many ECM-related genes are upregulated in SSAT (Figure 5a). Two significant proteoglycans in the ECM, VCAN and FMOD [17,18,19], are most abundant in SSAT (Figure 5a). Thus, the ECM is abundant in the SSAT depot, which results in a lower content of TG and crude fat. Meanwhile, the relaxed environment in the SSAT depot makes the fat deposition easier. Therefore, SSAT is abundant in river buffaloes. In addition, the cell morphology was incomplete in SSAT in the paraffin section while it was normal in other depots, which was partly due to the high content of the ECM. The ECM contains a large amount of collagen and proteoglycan, which form a highly ordered supra-macromolecular structure. As result, the ECM plays a vital role in nutrition and the protection of cells [12]. In addition, the ECM has many signal molecules that actively participate in the control of cell growth, polarity, shape, migration, and metabolic activities [13]. These results indicate that SSAT is not a simple AT, and biological functions need to be further explored.

AT is an important source of energy in the human diet. AT can have positive or negative effects on human health depending on FA composition [20]. It is well known that UFAs have a positive effect on human health, whereas SFAs do not. FA composition in various AT depots can be different [8,9,21]. Compared with VAT depots, SAT depots generally have a higher UFA content and a lower SFA content [8,9,16,21]. Our data also show that the UFA content in SAT depots (SSAT, BSAT, and IAT) is higher than that in VAT depots (OAT, PCAT, and PRAT) (Figure 2i). The UFA content is lower than the SFA content in all AT depots except for the SSAT depot (Figure 2j,k). The difference between AT depots in FA content is further supported by WGCNA. Both SAT (especially the SSAT depot) and UFA content had a positive correlation with yellow and green-yellow modules (Figure 4d). By contrast, both VAT (especially PRAT or PCAT depot) and SFA have a significant positive correlation with four modules, especially the brown module (Figure 4d). The brown module is significantly associated with FA β-oxidation (Figure 4g). Thus, the high UFA content in SSAT may be attributed to the abundant ECM, which protects FA from β-oxidation and results in high UFA accumulation in SSAT. Previous studies have indicated the metabolic protection of the ECM [10,11]. These results indicate that SSAT is more beneficial for human health than ATs in other depots.

### 3.3. PRAT Is Active in FA Oxidation for Energy Supply

As previously mentioned, excess energy can be stored in adipocytes mainly in TG droplets [16]. Our data indicate that PRAT is more active in FA β-oxidation, FA degradation, and metabolism (Figure 4f,g). TGs can be used for lipolysis to produce free FAs in the cytoplasm. Then, free FAs combine with CoA to form FA-CoA, which can be transported into mitochondria for oxidative decomposition to produce ATP [22]. Correspondingly, a large number of categories of mitochondria, tricarboxylic acid cycle (TAC cycle), and ATP were identified in brown and blue modules and were significantly associated with PRAT (Figure 4d,f,g). The heat shock protein 60 (*HSP60*, also known as *HSPD1*), a vital gene in maintaining mitochondrial morphology and regulating the oxidative phosphorylation in mitochondria [23,24], is the most abundant gene expressed in ATs and is significantly upregulated in PRAT (Figure 5d,e). In addition, the PRAT depot has a smaller cell size, which also indicates that TGs stored in here are more active in lipolysis and oxidation.

A previous study indicated that beige adipocytes are more active in VAT depots, especially in PRAT [25]. One pathway, thermogenesis (bta04714), was enriched in the brown module (Figure 4f), which had the most significant association with PRAT, followed by OAT (Figure 4d). Some known thermogenesis-associated genes (also known as brown or beige adipocyte markers), such as *UCP1*, *PRDM16*, and *TMEM26* [26,27], are expressed in VATs (Appendix A). In addition, hormone-sensitive lipase (*LIPE*, also known as *HSL*), a rate-limiting enzyme for lipolysis [28], had the highest expression level in PRAT (Appendix A). Therefore, our data support that PRAT is more active in energy metabolism. Nevertheless, more evidence is necessary to support this conclusion, as lipid oxidation and the number of mitochondria are generally limited in ATs.

### 3.4. Characterizing Adipose Tissue into Subcutaneous Fat and Visceral Fat Is Not Enough

Previous studies on transcriptional comparisons of AT depots have indicated that SAT is mainly associated with the ECM whereas VAT is closely linked to immunity in pigs [10,29], humans, and rodents [12,30]. In the present study, a comparison between SAT and VAT also supports this view (Figure 3e,f). However, further research indicates that dividing ATs into SAT and VAT is not enough [12] and ATs in different discrete locations should be considered as separate mini-organs [11,31]. In this study, WGCNA reveals two modules (yellow and green-yellow) associated with the ECM (Figure 4e, Appendix A). Both yellow and green-yellow modules have a high correlation with SAT (Figure 4d), consistent with previous studies [10,12,29,30]. However, a high positive correlation was only identified with the SSAT depot. Correlations between yellow and green-yellow modules and the other two SAT depots (BSAT and IAT) were low. Four modules (brown, blue, pink, and turquoise) had a significant positive correlation with VAT (*p* <0.05), but few items associated with immunity were identified in the turquoise module (Appendix A). The turquoise module had variable correlations with the three VAT depots (OAT, PCAT, and PRAT). The highest correlation was identified with OAT (Figure 4d), supporting the likelihood that OAT has a stronger inflammatory response than other VAT depots [10]. All these results support that ATs in multiple depots are different from each other in biological function.

Our findings support that ATs in different depots have variable origins. The homeobox genes are well known to mediate the major transitions in the metazoan body plan [32]. Based on all the identified genes, the six AT depots cannot be distinguished from each other in the present study (Figure 3c). Interestingly, the 26 identified homeobox genes divided the 6 AT depots into 4 clusters (Figure 3g). The visceral PRAT is close to the subcutaneous BSAT and IAT while it is distinguishable from the visceral OAT and PCAT (Figure 3g). Similarly, a recent study showed that the visceral retroperitoneal AT was distinguishable from the other two VATs (including the OAT) that are clustered more closely with SATs in pigs [10]. Our data indicate that the subcutaneous SSAT is distinguishable from the congeneric BSAT and IAT (Figure 3g), which is not consistent with that in pigs [10]. The visceral PCAT is the most dissimilar from the other five depots (Figure 3g), suggesting that PCAT is different from other depots in origin. These results support the likelihood that ATs in multiple discrete locations originate from different progenitor lines during development [33,34].

In conclusion, this study characterizes the cellular and molecular properties and the function and metabolism features of ATs from six discrete locations in river buffaloes. The present study further supports some previous views and has new findings, which are significant for the regulation of fat deposition and provide new insights into the features of AT depots in multiple discrete locations.

## 4. Materials and Methods

### 4.1. Sample Preparation

A total of 6 river buffaloes (approximately 20 months old) were used. All animals were raised in the Buffalo Research Institute of the Chinese Academy of Agricultural Sciences (Nanning, Guangxi, China) according to a standard feeding and management procedure. Briefly, calves were mainly fed with milk until 3 months old. From 4 to 6 months old, calves were mainly fed with concentrated feed and a small amount of high-quality coarse feed. From 7 to 20 months old, buffaloes were mainly fed with coarse feed and supplemented with concentrated feed. Male and female individuals were raised separately at 10 months old. Details on the age, breed, sex, and weight are presented in Appendix A. For each individual, ATs in six depots were sampled, including three SAT depots (BSAT, SSAT, and IAT) and three VAT depots (OAT, PCAT, and PRAT). The BSAT was sampled between 12th and 13th ribs. All the samples were immediately obtained after slaughter. For histology analysis, fresh adipose (1 cm × 0.5 cm × 0.5 cm) was fixed with 4% formaldehyde. For other assays, adipose tissue was stored in liquid nitrogen.

### 4.2. Histology of Adipose Tissue

All the 36 AT samples were used for histology analysis according to a previous study [25]. Briefly, the fixed sample was embedded in paraffin, sliced at a 4 µm thickness, stained with H&E, and imaged with a Nikon Eclipse Ci series microscope equipped. For each sample, three visual fields were captured for cell counting by ImageJ 1.48v software.

### 4.3. Triglyceride Measurements

The TG content of each sample was determined by a Triglyceride Kit (Applygen Technologies Inc., Beijing, China) according to the manufacturer’s protocol [35], including sample preparation and lysis and TG determination. For each sample, a total of 50 mg of tissue was used.

### 4.4. Fatty Acid Analysis

Crude fat was extracted from AT using an acid hydrolysis–petroleum ether extraction method. For each sample, a total of 60 mg of crude fat was dissolved by 5 mL of isooctane. Then, 200 μL of methanol potassium hydroxide was added and shaken violently for 30 s. When the liquid became clear, 1 g of sodium bisulfate was added and shaken again to neutralize the potassium hydroxide. Then, the supernatant was transferred to an auto-sample vial after salt precipitation. The fatty acid methyl ester was analyzed using a gas chromatograph (GC6890, Agilent, HP, USA). The reference standard was purchased from ANPEL. The percentage of a certain fatty acid was calculated according to the following formula:$$Yi=\frac{\mathrm{As}\times F_{{\mathrm{FAME}}_{i}-FA_{i}}}{\sum A_{Si}\times F_{{\mathrm{FAME}}_{i}-FA_{i}}}$$

*Y_i_*—the percentage of a certain fatty acid in the sample, %;

*A_Si_*—peak area of each fatty acid methyl ester in the sample;

$F_{{\mathrm{FAME}}_{i}-FA_{i}}$—conversion coefficient of fatty acid methyl ester into fatty acid;

∑*A_Si_*—sum of peak areas of all the fatty acid methyl esters in the sample.

### 4.5. RNA Isolation and Sequencing

In total, 36 ATs (6 depots × 6 buffaloes) were used for RNA sequencing. The total RNA was isolated using the RNeasy Lipid Tissue Mini Kit (QIAGEN, Hilden, Germany) according to the manual instruction. An rRNA-depleted library for each sample was constructed and sequenced by the Illumina HiSeq4000 platform (Illumina, San Diego, CA, USA).

### 4.6. Transcriptome Assembly and mRNA Expression Analysis

Raw data were filtered with Fastp (v0.20.1) [36] to remove reads with low quality or adapters. Clean data were mapped to a river buffalo genome with HISAT2 (v2.2.1) [37]. The river buffalo genome was assembled by our team (not published) and can be provided upon request. Mapped reads were assembled by StringTie (v2.1.4) [38] to obtain transcripts. The assembled transcript was annotated by Gffcompare (v0.11.2) [39]. Expression of the transcript was analyzed by StringTie (v2.1.4) [38] and presented as fragments per kilobase of transcript per million mapped reads (FPKM). All the parameters were set to default.

### 4.7. Differential Expression Analysis

Transcripts with an average of FPKM ≥10 were used for differential expression analysis by DESeq2 (v1.32.1) [40]. A *p*-value <0.05 and log2(fold change) ≥1.5 were used as cut-offs for statistical significance.

### 4.8. Weighted Gene Co-Expression Network Analysis

The weighted gene co-expression network analysis (WGCNA) was performed by the R package “WGCNA” with a one-step method [41]. Transcripts with average FPKM ≥2 (approximately 10,100 transcripts) were used for WGCNA, with the number of transcripts within a module ≥30 and the combination coefficient = 0.25. The correlation between phenotype and gene expression was examined by the labeled Heatmap function to obtain the correlated co-expression modules.

### 4.9. Functional Enrichment Analysis

Functional enrichment analysis of Gene Ontology (GO) terms and the KEGG pathway was performed using the online database DAVID. A benjamini value <0.05 was considered statistically significant.

### 4.10. Statistical Analysis

Normal distribution testing was performed for the data. One-way ANOVA was applied for comparison by SPSS software 19.0 (IBM, Armonk, NY, USA). A *p*-value < 0.05 was considered statistically significant.

## Figures and Tables

**Figure 1 ijms-24-08410-f001:**
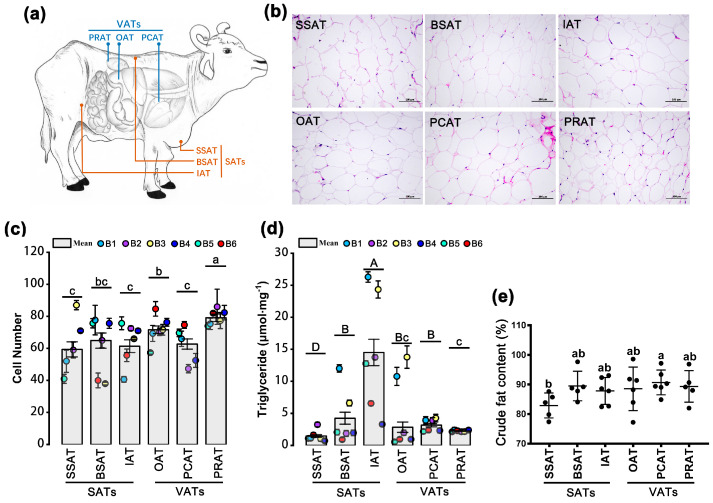
Characteristic ATs in the six depots. (**a**) Sources of tissues: three subcutaneous adipose tissues (SATs) (back subcutaneous adipose tissues, BSATs; sternum subcutaneous adipose tissues, SSATs; inguinal adipose tissue, IATs) and three visceral adipose tissues (VATs) (omental adipose tissue, OAT; pericardial adipose tissue, PCAT; perirenal adipose tissue, PRAT). (**b**) H&E staining of adipose tissues in the six depots. The difference in cell number (**c**), triglyceride content (**d**), and crude fat content (**e**). Crude fat content is presented as a percentage by weight, namely the extracted weight (g) of crude fat from 100 g adipose tissue. A total of six buffaloes (numbered B1, B2, B3, B4, B5, and B6) were used. Three views for each independent depot were captured for cell counting. Three samples for each independent depot were used for triglyceride detection. Data are presented as mean ± SE. An uppercase letter indicates *p* < 0.01; lowercase letters indicate *p* < 0.05.

**Figure 2 ijms-24-08410-f002:**
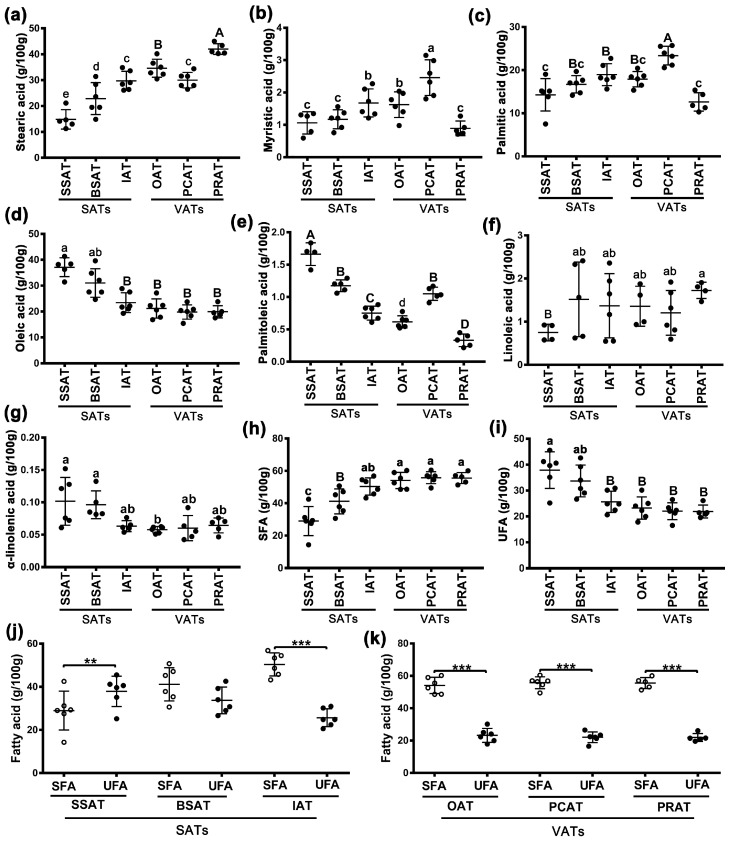
Fatty acid composition in the six adipose depots: three saturated fatty acids (SFAs), including stearic acid (**a**), myristic acid (**b**), and palmitic acid (**c**), and four unsaturated fatty acids (UFAs), including oleic acid (**d**), palmitoleic acid (**e**), linoleic acid (**f**), and α-linolenic acid (**g**). The difference in SFA (**h**) and UFA (**i**) contents among adipose tissue depots. Composition of SAF and UFA in subcutaneous adipose depots (**j**) and visceral adipose depots (**k**). Lowercase letter indicates *p* < 0.05; uppercase letter and ** indicates *p* < 0.01; *** indicates *p* < 0.001. circles and solid circles indicate *n* = 6.

**Figure 3 ijms-24-08410-f003:**
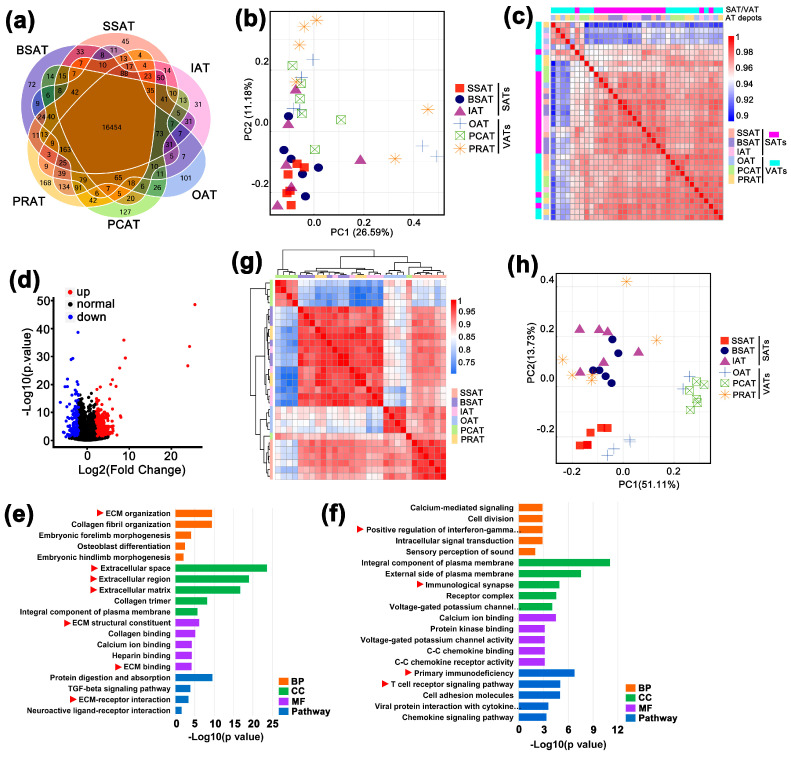
Transcriptional features and functional divergence between SATs and VATs. (**a**) A Venn chart for all the transcripts identified in the six adipose depots. (**b**) Scatter plot of the first two principal component vectors and (**c**) hierarchical clustering using all the transcripts identified in the 36 AT samples from the 6 depots. (**d**) Differentially expressed genes between SATs and VATs. Blue spots indicate downregulated genes in SATs while red spots indicate upregulated genes in VATs. Representation of functional enrichment of genes upregulated in SATs (**e**) and VATs (**f**). Only the top 5 items are presented for each category. Red triangles indicate those items associated with extracellular matrix (**e**) or immune system (**f**). (**g**) Hierarchical clustering and scatter plot of the first 2 principal component vectors (**h**) using the 26 HOX paralogs identified in the transcriptome.

**Figure 4 ijms-24-08410-f004:**
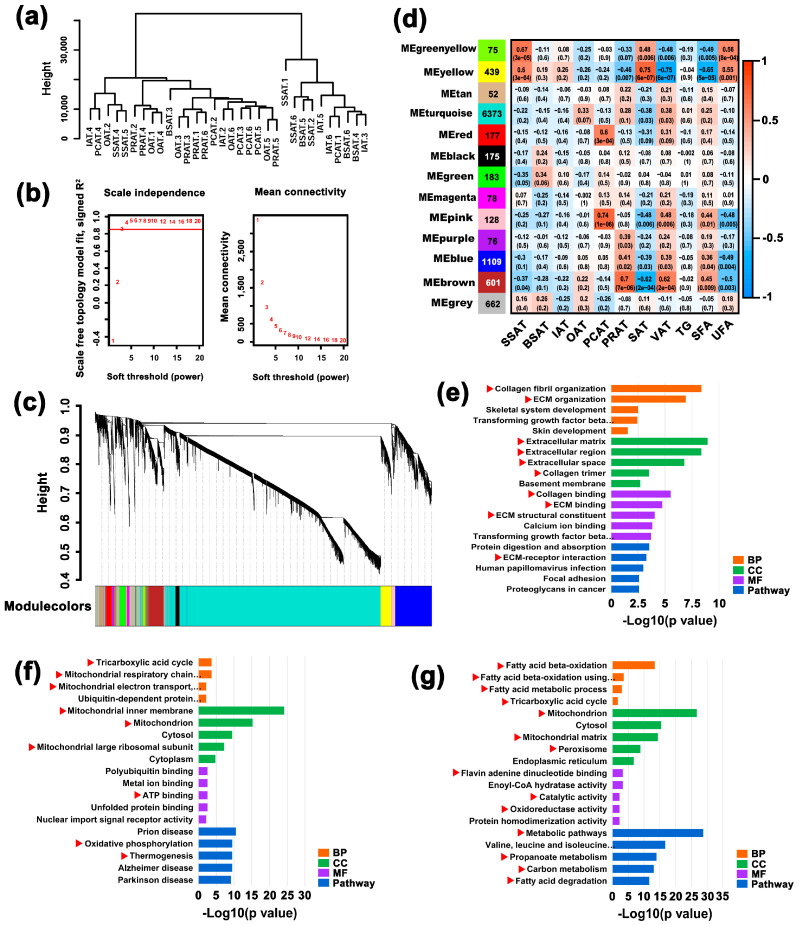
WGCNA reveals functional divergence among different AT depots. (**a**) Cluster analysis using transcripts with average FPKM ≥2 across the 32 AT samples. Four samples deviating from others were removed. (**b**) Network topology for various soft-thresholding powers. (**c**) Gene dendrogram clustering the dissimilarity based on the consensus topological overlap. Modules are indicated by the color row. (**d**) Relationships between modules and traits. Each cell presents the correlation coefficient between module (row) and trait (column) and *p*-value (in parentheses). Modules are indicated by the color column and the number in the corresponding color indicates the number of genes involved in the corresponding module. (**e**) Functional enrichment by genes involved in yellow and green-yellow modules. The top 5 items are presented for each category. Red triangles indicate those items associated with the extracellular matrix. (**f**) Functional enrichment by genes involved in the brown module. The top 5 items are presented for each category. Red triangles indicate those items associated with the tricarboxylic acid cycle, mitochondria, and ATP. (**g**) Functional enrichment by genes involved in the blue module. The top 5 items are presented for each category. Red triangles indicate those items associated with FA beta-oxidation, degradation and metabolism, and mitochondria.

**Figure 5 ijms-24-08410-f005:**
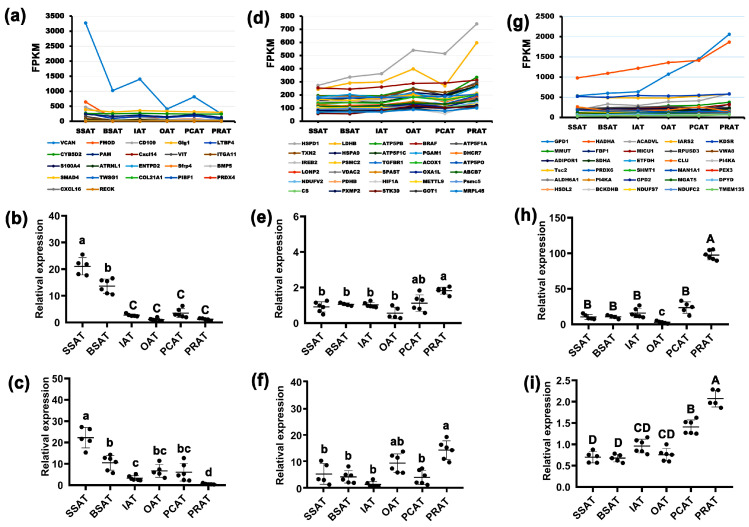
Expression profiles of candidate genes. (**a**) Expressional profiles of genes involved in the candidate items in yellow and green-yellow modules across the 6 AT depots by RNA sequencing. Expressional profiles of *VCAN* (**b**) and *FMOD* (**c**) across the 6 AT depots by qRT-PCR. (**d**) Expressional profiles of genes involved in the candidate items in brown module across the 6 AT depots by RNA sequencing. Only genes with high expression in PRAT (top 30) are presented. Expressional profiles of *HSPD1* (**e**) and *LDHB* (**f**) across the 6 AT depots by qRT-PCR. (**g**) Expressional profiles of genes involved in the candidate items in blue module across the 6 AT depots by RNA sequencing. Only genes with high expression in PRAT (top 30) are presented. Expressional profiles of *GPD1* (**h**) and *HADHA* (**i**) across the 6 AT depots by qRT-PCR. Data are presented as means ± SE. Uppercase letter indicates *p* < 0.01; lowercase letter indicates *p* < 0.05.

## Data Availability

The RNA sequencing data were deposited in the Genome Expression Omnibus (GEO) of NCBI (https://www.ncbi.nlm.nih.gov/geo/, accessed on 19 July 2022). The accession number is GSE208541.

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
