# Peer review of "Diversity in Cell Morphology, Composition, and Function among Adipose Depots in River Buffaloes"

_ijms, 2023, doi:10.3390/ijms24098410_

Round 1
Reviewer 1 Report
The manuscript entitled " Diversity in cell morphology, composition and function among adipose depots in river buffaloes" by Yang, et al., is a very interesting article that explains how cell size, composition, and function of six adipose tissue depots in river buffaloes. This study provides cellular and molecular features, metabolism, and origin of adipose tissue depots in buffaloes. Overall, a well-written manuscript; however, a few minor points should be addressed.
Comments:
1. Is there a gender difference in the six adipose tissue depots in river buffaloes? Though male and female buffalo were mentioned in sampling, there is no discussion about gender differences in the manuscript.
2. Line 273: “PRAT had a significantly high correlation with FA beta-oxidation and degradation, tricarboxylic acid cycle, and energy metabolism (Figure 4d-4f)” why did the author exclude beta-oxidation - 4G?
3. Line 303: “The UFA content is lower than the SFA content in all AT depots except for the SSAT depot (Figure 3j)” doesn’t match the results described.
4. Figures 5A, 5D, & 5G are fuzzy and not clear update with high-resolution images.
Author Response
The manuscript entitled "Diversity in cell morphology, composition and function among adipose depots in river buffaloes" by Yang, et al., is a very interesting article that explains how cell size, composition, and function of six adipose tissue depots in river buffaloes. This study provides cellular and molecular features, metabolism, and origin of adipose tissue depots in buffaloes. Overall, a well-written manuscript; however, a few minor points should be addressed.
Comments:
- Is there a gender difference in the six adipose tissue depots in river buffaloes? Though male and female buffalo were mentioned in sampling, there is no discussion about gender differences in the manuscript.
Answer: Thank you for your careful and professional review. As adipose development is affected by gender, we also considered the gender when we designed the study. However, no significant difference was found in phenotype (cell size and composition) between male and female individuals. Thus, the samples were not grouped according to the gender. At last, some interesting results were revealed by ignoring the gender, indicating that these results were common in both male and female individuals. We further discuss this issue in lines 259-264.
- Line 273: “PRAT had a significantly high correlation with FA beta-oxidation and degradation, tricarboxylic acid cycle, and energy metabolism (Figure 4d-4f)” why did the author exclude beta-oxidation - 4G?
Answer: Thank you for your careful review. We feel sorry that we had make such a mistake. It should be “Figure 4d,g” in line 279.
- Line 303: “The UFA content is lower than the SFA content in all AT depots except for the SSAT depot (Figure 3j)” doesn’t match the results described.
Answer: Thank you for your kind comment. It should be “Figure 2j,k” in line 309. We feel sorry that we had make such a mistake. Previously, figures and manuscript had undergone many revisions.
4. Figures 5A, 5D, & 5G are fuzzy and not clear update with high-resolution images.
Answer: Thank you for your kind suggestion. In the official version, a clear, high-quality file will be submitted alone.
Reviewer 2 Report
In this manuscript, differences in morphology, fatty acid profile and transcriptome in subcutaneous SAT) and visceral adipose tissue (VAT) in river buffaloes were investigated. The authors found that subcutaneous fat tissue has larger adipocytes, a higher concentration of unsaturated fatty acids and more extracellular matrix. Transcriptional comparisons of AT depots have confirmed earlier findings about the differences between SAT and VAT, and furthermore showed differences in AT depending on localization in the body. The authors conclude that SAT je associated with extracellular matrix, whereas VAT is involved with immunity
The conclusions of this study cannot be accepted for several reasons:
The biggest shortcoming of the study is that the authors classified palmitic acid (FA 16:0) among monounsaturated FAs, while palmitoleic acid (FA 16:1 n-7) among saturated acids. This is not just a name confusion, as the determined concentrations correctly correspond to the FA names. Palmitic acid, the most common saturated FA found in human and animals, is ingested as part of the diet or can be produced by de novo lipid synthesis from carbohydrate consumption. Because it is in high concentration, it greatly affects the obtained data and their interpretation. On the other hand, palmitoleic acid is a monounsaturated FA, which is relatively non-toxic to cells and protect cells against saturated FA-induced lipotoxicity by redirecting toxic saturated FA into triglyceride storage.
Because misclassification of FA is used throughout the study, it invalidates all results, including
transcriptional analyses and functional divergence between SATs and VATs in different locations.
Supplementary Materials was not available
e in the review offer.
Figure 1: Is the definition of "crude fat content" missing or why is it expressed as a % of what??
The fact that the subcutaneous AT has more extracellular matrix - connective tissue, was demonstrated many years ago and is connected to its function as a supporting tissue.
In the methodological section, the method used for extracting triglycerides from the tissue, before their determination with the kit, is not described. and the reference of this laborious analysis is not given.
Line 104: the correct name for linolenic acid is α-linolenic acid. It cannot be written without alpha.
Line 36: We cannot agree with the explanation that the reason for the discrepancy between TG concentration and crude fat content in retroperitoneal AT is a consequence of changes in β-oxidation. In AT, changes in TG are a consequence of lipolysis, which provides release of FA for transport to muscle and liver for β-oxidation. See also line 329.
Line 329: Lipid oxidation in AT, as well as the number of mitochondria, are extremely small, providing only the metabolic requirements for adipocyte viability. β-oxidation takes place in muscle tissue, myocardium and others.
Line 366: The results are also inconsistent with findings in laboratory rats, where retroperitoneal AT has accentuated TG storage, greater cell hypertrophy, and lower metabolic activity compared to epididymal or mesenteric AT.
Overall, it is not clear whether the focus of the study is interesting for readers. As a herbivore, the buffalo has a completely different metabolism from humans, and its AT is a waste product. This study also does not address the problem of increasing fat content in muscles to improve meat quality. In river buffaloes, the muscle tissue contains very little lipids, which is the reason for the low quality of the meat for consumption. The opposite is the case in humans, where ectopic accumulation of lipids in muscle tissue leads to insulin resistance and increases the development of metabolic syndrome, type 2 diabetes even without the presence of obesity.
Author Response
In this manuscript, differences in morphology, fatty acid profile and transcriptome in subcutaneous SAT) and visceral adipose tissue (VAT) in river buffaloes were investigated. The authors found that subcutaneous fat tissue has larger adipocytes, a higher concentration of unsaturated fatty acids and more extracellular matrix. Transcriptional comparisons of AT depots have confirmed earlier findings about the differences between SAT and VAT, and furthermore showed differences in AT depending on localization in the body. The authors conclude that SAT associated with extracellular matrix, whereas VAT is involved with immunity.
The conclusions of this study cannot be accepted for several reasons:
The biggest shortcoming of the study is that the authors classified palmitic acid (FA 16:0) among monounsaturated FAs, while palmitoleic acid (FA 16:1 n-7) among saturated acids. This is not just a name confusion, as the determined concentrations correctly correspond to the FA names. Palmitic acid, the most common saturated FA found in human and animals, is ingested as part of the diet or can be produced by de novo lipid synthesis from carbohydrate consumption. Because it is in high concentration, it greatly affects the obtained data and their interpretation. On the other hand, palmitoleic acid is a monounsaturated FA, which is relatively non-toxic to cells and protect cells against saturated FA-induced lipotoxicity by redirecting toxic saturated FA into triglyceride storage.
Because misclassification of FA is used throughout the study, it invalidates all results, including transcriptional analyses and functional divergence between SATs and VATs in different locations.
Answer: Thank you for your careful review and professional comments and suggestions. We feel very sorry for such a mistake. This is a critical issue. We then checked the raw data carefully. We now confirm that (1) the names of palmitic acid and palmitoleic acid were confused; (2) figure 2c and 2e were misused; (3) nevertheless, the results of total SFA and UFA are correct for all the analysis was initially performed based on their Chinese names. In the new version, we have (1) corrected the names of palmitic acid and palmitoleic acid throughout the manuscript; (2) interchanged the positions of figure 2c and 2e; (3) rewritten the results corresponding to figure 2c and 2e in lines 109-112. The raw data can be provided if it is necessary. We are very sorry for presenting such a bad impression. Importantly, we are very grateful for your professional comments and suggestions.
Supplementary Materials was not available in the review offer.
Answer: Thank you for your kind comment. We have submitted the supplementary materials in the new version.
Figure 1: Is the definition of "crude fat content" missing or why is it expressed as a % of what??
Answer: Thank you for your careful review. We feel sorry for such a mistake. It should be (%), not (100%) in Figure 1e, which has been updated. It is a percentage by weight, namely the extracted weight (g) of crude fat from 1g adipose tissue. The definition of "crude fat content" has been added in lines 87-88.
The fact that the subcutaneous AT has more extracellular matrix - connective tissue, was demonstrated many years ago and is connected to its function as a supporting tissue.
Answer: Thank you for your professional comments. Data in the present study also support this conclusion, indicating the accuracy and reliability of the sample and analysis in the present study. In addition, results obtained in buffaloes are consistent with those in other mammals, indicating that results obtained in buffaloes have certain reference value for other mammals.
In the methodological section, the method used for extracting triglycerides from the tissue, before their determination with the kit, is not described. and the reference of this laborious analysis is not given.
Answer: Thank you for your professional suggestions. The manufacturer's instructions provide a detailed introduction to the entire process of sample preparation and lysis and TG determination. According to the manufacturer's instructions, 50 mg fresh tissue was accurately weighed and fully ground in a mortar. This instruction has been added in lines 402-404. The kit method for TG determination of tissues has been widely used. Thus, details were not provided in the present study and the reference.
Line 104: the correct name for linolenic acid is α-linolenic acid. It cannot be written without alpha.
Answer: Thank you for your professional suggestions. We are sorry for such a mistake. It has been corrected in lines 104 and 121 and figure 2g.
Line 36: We cannot agree with the explanation that the reason for the discrepancy between TG concentration and crude fat content in retroperitoneal AT is a consequence of changes in β-oxidation. In AT, changes in TG are a consequence of lipolysis, which provides release of FA for transport to muscle and liver for β-oxidation. See also line 329.
Answer: Thank you for your professional comments. We compared the results of crude fat and TG content again. Except for IAT, the results of TG content are mainly consistent with those of crude fat content. Thus, we remove the inappropriate discussion. An unusually high content of TG in IAT confused us. We even re-detected the TG content in IAT for several times. In addition, WGCNA analysis failed to connect TG content with IAT. Thus, we did not discuss this issue that why IAT had an unusually high content of TG.
Line 329: Lipid oxidation in AT, as well as the number of mitochondria, are extremely small, providing only the metabolic requirements for adipocyte viability. β-oxidation takes place in muscle tissue, myocardium and others.
Answer: Thank you for your professional and kind comments. Data in the present study support that β-oxidation, FA degradation, and metabolism are more active in perirenal adipose tissue (PRAT) than other depots in river buffaloes. We were also very surprised by this result at the beginning. Perhaps this is one of the special aspects of PRAT. Nevertheless, more evidence is necessary to support this conclusion. Thus, a sentence has been further added in lines 341-342.
Line 366: The results are also inconsistent with findings in laboratory rats, where retroperitoneal AT has accentuated TG storage, greater cell hypertrophy, and lower metabolic activity compared to epididymal or mesenteric AT.
Answer: Thank you for your professional and kind comments. Retroperitoneal AT was not included in the present study. In the present study, three subcutaneous AT depots (back subcutaneous AT, BSAT; sternum subcutaneous AT, SSAT; inguinal AT, IAT) and three visceral AT depots (omental AT, OAT; pericardial AT, PCAT; perirenal AT, PRAT) in were studied. As in my opinion, the findings in laboratory rats are consistent with the results in the present study. In laboratory rats, retroperitoneal AT has accentuated TG storage and greater cell hypertrophy but it has a lower metabolic activity. In the present study, perirenal AT (PRAT) has a smaller cell size but it has a high metabolic activity.
Overall, it is not clear whether the focus of the study is interesting for readers. As an herbivore, the buffalo has a completely different metabolism from humans, and its AT is a waste product. This study also does not address the problem of increasing fat content in muscles to improve meat quality. In river buffaloes, the muscle tissue contains very little lipids, which is the reason for the low quality of the meat for consumption. The opposite is the case in humans, where ectopic accumulation of lipids in muscle tissue leads to insulin resistance and increases the development of metabolic syndrome, type 2 diabetes even without the presence of obesity.
Answer: Thank you very much for your careful review and professional comments and suggestions. The present study enhances our understanding of the cellular and molecular features, metabolism, and origin of AT depots in buffaloes, which is significant for the regulation of fat deposition and provides new insights into the features of AT depots in multiple discrete locations. Intramuscular AT is nearly impossible to be sampled in buffaloes. Thus, we cannot directly study the intramuscular AT depot. Instead, intramuscular preadipocytes can be isolated for research, which is what we are doing now. Although the aim of the regulation of fat deposition is different between humans and livestock animals, the developmental characteristics and biological functions of adipose tissues are similar between humans and livestock animals. Thus, we believe that the results revealed in buffaloes can also provide important information for the research in humans and other mammals.
Round 2
Reviewer 2 Report
The authors addressed all my comments and several improvements have been made to the manuscript. However, not all of my previous comments were adequately addressed,
It is a pity that the authors did not attach the file Supplements to the form for the assessment of the manuscript, i.e. for download. The authors did not understand my objection that the link to the supplement in the manuscript is not functional until the article is published.
To improve knowledge about adipose tissue, I remind you that the adipose tissue around the kidneys is called either perirenal AT or retroperitoneal AT. Both names mean the same thing.
Although some of my reservations about this manuscript remain, I believe it is now suitable for publication
The authors addressed all my comments and several improvements have been made to the manuscript. However, not all of my previous comments were adequately addressed,
It is a pity that the authors did not attach the file Supplements to the form for the assessment of the manuscript, i.e. for download. The authors did not understand my objection that the link to the supplement in the manuscript is not functional until the article is published.
To improve knowledge about adipose tissue, I remind you that the adipose tissue around the kidneys is called either perirenal AT or retroperitoneal AT. Both names mean the same thing.
Although some of my reservations about this manuscript remain, I believe it is now suitable for publication